Extended Abstract Track

# On the Expressive Power of
# Geometric Graph Neural Networks

**Chaitanya K. Joshi**[*,1]                                    CHAITANYA.JOSHI@CL.CAM.AC.UK
**Cristian Bodnar**[*,1]                                              CB2015@CAM.AC.UK
**Simon V. Mathis**[1]                                          SIMON.MATHIS@CL.CAM.AC.UK
**Taco Cohen**[2]                                              TACOS@QTI.QUALCOMM.COM
**Pietro Liò**[1]                                               PIETRO.LIO@CL.CAM.AC.UK

[1] *Department of Computer Science and Technology, University of Cambridge, UK*
[2] *Qualcomm AI Research, The Netherlands*[†]

**Editors:** Sophia Sanborn, Christian Shewmake, Simone Azeglio, Arianna Di Bernardo, Nina Miolane

## Abstract

We propose a geometric version of the Weisfeiler-Leman graph isomorphism test (GWL) for discriminating geometric graphs while respecting the underlying physical symmetries: permutations, rotation, reflection, and translation. We use GWL to characterise the expressive power of Graph Neural Networks (GNNs) that are invariant or equivariant to physical symmetries in terms of the classes of geometric graphs they can distinguish. This allows us to formalise the advantages of equivariant GNNs over invariant GNNs: equivariant layers have greater expressive power as they enable propagating geometric information beyond local neighbourhoods, while invariant layers only reason locally via scalars and cannot discriminate geometric graphs with different non-local properties.

**Keywords:** Geometric Deep Learning, Graph Neural Networks, Graph Isomorphism

## 1. Introduction

The graph isomorphism problem and the Weisfeiler-Leman (WL) (Weisfeiler and Leman, 1968) test for distinguishing non-isomorphic graphs have become a powerful tool for analysing the expressive power of Graph Neural Networks (GNNs) (Xu et al., 2019; Morris et al., 2019). The WL framework has been a major driver of progress for more expressive GNNs (Maron et al., 2019; Morris et al., 2020; Bodnar et al., 2021a). However, WL does not directly apply to the increasingly relevant special case of *geometric graphs* – graphs embedded in Euclidean space – which come equipped with a stronger notion of isomorphism that also takes spatial symmetries into account. The lack of theoretical tools is becoming more apparent as geometric graphs are increasingly used to model systems in biochemistry (Jamasb et al., 2022), material science (Chanussot et al., 2021), and multiagent robotics (Li et al., 2020). Graph Neural Networks (GNNs) with Euclidean symmetries 'baked in' have emerged as the architecture of choice for these domains (Geiger and Smidt, 2022).

Geometric GNNs follow the message passing paradigm (Gilmer et al., 2017) where node features are updated in a permutation equivariant manner by aggregating features from local neighbourhoods. In addition to the permutation group, the geometric attributes of the

---

[*] Equal first authors.

[†] Qualcomm AI Research is an initiative of Qualcomm Technologies, Inc.

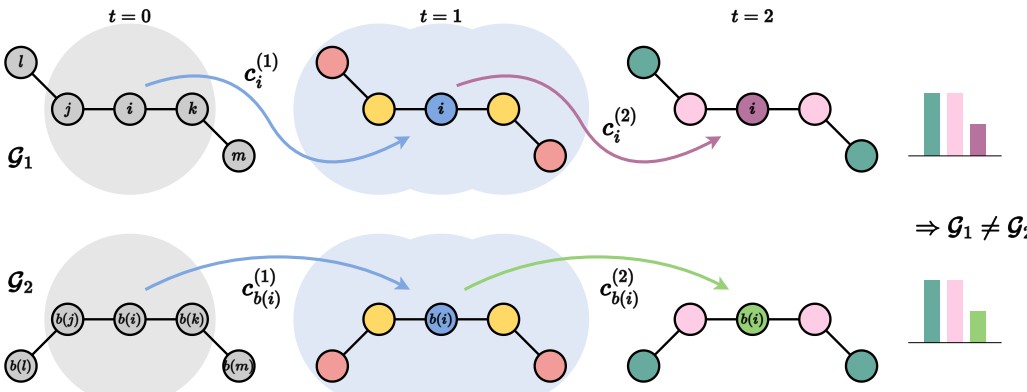

Figure 1: **Geometric Weisfeiler-Leman Test.** GWL distinguishes non-isomorphic geometric graphs $\mathcal{G}_1$ and $\mathcal{G}_2$ by injectively assigning colours to distinct neighbourhood patterns, up to global symmetries (here $\mathfrak{G} = O(d)$). Each iteration expands the neighbourhood from which geometric information can be gathered (shaded for node $i$). Example inspired by Schütt et al. (2021).

nodes (*e.g.* coordinates, velocity) transform along with Euclidean transformations of the system, *i.e.* they are equivariant to a Lie group such as the group of rotations ($SO(d)$) or rotations and reflections ($O(d)$). We use $\mathfrak{G}$ as a generic symbol for such a Lie group. Based on this, we consider two classes of GNNs for geometric graphs: (1) $\mathfrak{G}$-**equivariant models**, where the intermediate features and propagated messages are $\mathfrak{G}$-equivariant geometric quantities such as vectors or tensors (Thomas et al., 2018; Satorras et al., 2021); and (2) $\mathfrak{G}$-**invariant models**, which only propagate $\mathfrak{G}$-invariant scalar features such as distances and angles (Schütt et al., 2018; Gasteiger et al., 2020). Despite promising empirical results for both classes of architectures, key theoretical questions remain unanswered: (1) How to characterise the *expressive power* of geometric GNNs? And (2) what is the tradeoff between $\mathfrak{G}$-equivariant and $\mathfrak{G}$-invariant GNNs?

**Contributions.** In this work, we study the expressive power of geometric GNNs from the perspective of discriminating non-isomorphic geometric graphs. We propose a geometric version of the Weisfeiler-Leman graph isomorphism test, termed GWL. (Figure 1). We use GWL to formally characterise classes of graphs that can and cannot be distinguished by $\mathfrak{G}$-invariant and $\mathfrak{G}$-equivariant GNNs. We show how invariant models have limited expressive power as they only reason locally via scalar quantities, while equivariant models distinguish a larger class of graphs by propagating geometric vector quantities beyond local neighbourhoods.

For **Background and Preliminaries**, please see Appendix A.

## 2. The Geometric Weisfeiler-Leman Test

**Assumptions.** Analogously to the WL test, the geometric and scalar features the nodes are equipped with come from countable subsets $C \subset \mathbb{R}^d$ and $C' \subset \mathbb{R}$, respectively. As a

# Extended Abstract Track

result, when we require functions to be injective, we require them to be injective over the countable set of $\mathfrak{G}$-orbits that are obtained by acting with $\mathfrak{G}$ on the dataset.

**Intuition.** For an intuition of how to generalise the WL test to geometric graphs, we note that WL uses a local, node-centric, procedure to update the colour of each node $i$ using the colours of its 1-hop *neighbourhood* $\mathcal{N}_i$. In the geometric setting, $\mathcal{N}_i$ is an attributed point cloud around the central node $i$. As a result, each neighbourhood carries two types of information: (1) neighbourhood type (invariant to $\mathfrak{G}$) and (2) neighbourhood geometric orientation (equivariant to $\mathfrak{G}$). From an axiomatic point of view, our generalisation of the WL neighbourhood aggregation procedure must meet two properties:

**Property 1: Orbit injectivity of colours.** If two neighbourhoods are the same up to an action of $\mathfrak{G}$ (*e.g.* rotation), then the colours of the corresponding central nodes should be the same. Thus, the colouring must be $\mathfrak{G}$-orbit injective – which also makes it $\mathfrak{G}$-invariant – over the countable set of all orbits of neighbourhoods in our dataset.

**Property 2: Preservation of local geometry.** A key property of WL is that the aggregation is injective. A $\mathfrak{G}$-invariant colouring procedure that purely satisfies Property 1 is not sufficient because, by definition, it loses spatial properties of each neighbourhood such as the relative pose or orientation (Hinton et al., 2011). Thus, we must additionally update auxiliary *geometric information* variables in a way that is $\mathfrak{G}$-equivariant and injective.

**Geometric Weisfeiler-Leman (GWL).** These intuitions motivate the following definition of the GWL test. At initialisation, we assign to each node $i \in \mathcal{V}$ a scalar node colour $c_i \in C'$ and an auxiliary object $\boldsymbol{g}_i$ containing the geometric information associated to it:

$$c_i^{(0)} := \text{HASH}(\boldsymbol{s}_i), \quad \boldsymbol{g}_i^{(0)} := \left( c_i^{(0)}, \overrightarrow{\boldsymbol{v}}_i \right), \tag{1}$$

where HASH denotes an injective map over the scalar attributes $\boldsymbol{s}_i$ of node $i$. To define the inductive step, assume we have the colours of the nodes and the associated geometric objects at iteration $t - 1$. Then, we can aggregate the geometric information around node $i$ into a new object as follows:

$$\boldsymbol{g}_i^{(t)} := \left( (c_i^{(t-1)}, \boldsymbol{g}_i^{(t-1)}), \ \{\!\{(c_j^{(t-1)}, \boldsymbol{g}_j^{(t-1)}, \overrightarrow{\boldsymbol{x}}_{ij}) \mid j \in \mathcal{N}_i\}\!\} \right), \tag{2}$$

Importantly, the group $\mathfrak{G}$ can act on the geometric objects above inductively by acting on the geometric information inside it. This amounts to rotating (or reflecting) the entire $t$-hop neighbourhood contained inside:

$$\mathfrak{g} \cdot \boldsymbol{g}_i^{(0)} := \left( c_i^{(0)}, \ \boldsymbol{Q}_{\mathfrak{g}} \overrightarrow{\boldsymbol{v}}_i \right), \quad \mathfrak{g} \cdot \boldsymbol{g}_i^{(t)} := \left( (c_i^{(t-1)}, \mathfrak{g} \cdot \boldsymbol{g}_i^{(t-1)}), \ \{\!\{(c_j^{(t-1)}, \mathfrak{g} \cdot \boldsymbol{g}_j^{(t-1)}, \boldsymbol{Q}_{\mathfrak{g}} \overrightarrow{\boldsymbol{x}}_{ij}) \mid j \in \mathcal{N}_i\}\!\} \right)$$

Clearly, the aggregation building $\boldsymbol{g}_i$ for any time-step $t$ is injective and $\mathfrak{G}$-equivariant. Finally, we can compute the node colours at iteration $t$ for all $i \in \mathcal{V}$ by aggregating the geometric information in the neighbourhood around node $i$:

$$c_i^{(t)} := \text{I-HASH}^{(t)}\left( \boldsymbol{g}_i^{(t)} \right), \tag{3}$$

by using a $\mathfrak{G}$-orbit injective and $\mathfrak{G}$-invariant function that we denote by I-HASH. That is for any geometric objects $\boldsymbol{g}, \boldsymbol{g}'$, I-HASH$(\boldsymbol{g}) = $ I-HASH$(\boldsymbol{g}')$ if and only if there exists $\mathfrak{g} \in \mathfrak{G}$ such that $\boldsymbol{g} = \mathfrak{g} \cdot \boldsymbol{g}'$.

Joshi Bodnar Mathis Cohen Liò

# *Extended Abstract Track*

**Overview.** With each iteration, $\boldsymbol{g}_i^{(t)}$ aggregates geometric information in progressively larger $t$-hop subgraph neighbourhoods $\mathcal{N}_i^{(t)}$ around the node $i$. The node colours summarise the structure of these $t$-hops via the $\mathfrak{G}$-invariant aggregation performed by I-HASH. The procedure terminates when the partitions of the nodes induced by the colours do not change from the previous iteration. Finally, given two geometric graphs $\mathcal{G}$ and $\mathcal{H}$, if there exists some iteration $t$ for which $\{\!\{c_i^{(t)} \mid i \in \mathcal{V}(\mathcal{G})\}\!\} \neq \{\!\{c_i^{(t)} \mid i \in \mathcal{V}(\mathcal{H})\}\!\}$, then GWL deems the two graphs as being geometrically non-isomorphic. Otherwise, we say the test cannot distinguish the two graphs.

**Invariant GWL.** Since we are interested in understanding the role of $\mathfrak{G}$-equivariance, we also consider a more restrictive Invariant GWL (IGWL) that only updates node colours using the $\mathfrak{G}$-orbit injective I-HASH function and does not propagate geometric information:

$$c_i^{(t)} := \text{I-HASH}\left((c_i^{(t-1)}, \overrightarrow{\boldsymbol{v}}_i) , \{\!\{(c_j^{(t-1)}, \overrightarrow{\boldsymbol{v}}_j, \overrightarrow{\boldsymbol{x}}_{ij}) \mid j \in \mathcal{N}_i\}\!\}\right). \tag{4}$$

## 2.1. What Geometric Graphs can GWL and IGWL Distinguish?

In order to formalise the expressive power of GWL and IGWL, let us consider what geometric graphs can and cannot be distinguished by the tests. As a simple first observation, we note that when all coordinates and vectors are set equal to zero GWL coincides with the standard 1-WL. In this *edge case*, GWL has the same expressive power as 1-WL.

Next, let us consider consider the simplified setting of two geometric graphs $\mathcal{G}_1 = (\boldsymbol{A}_1, \boldsymbol{S}_1, \overrightarrow{\boldsymbol{V}}_1, \overrightarrow{\boldsymbol{X}}_1)$ and $\mathcal{G}_2 = (\boldsymbol{A}_2, \boldsymbol{S}_2, \overrightarrow{\boldsymbol{V}}_2, \overrightarrow{\boldsymbol{X}}_2)$ such that the underlying attributed graphs $(\boldsymbol{A}_1, \boldsymbol{S}_1)$ and $(\boldsymbol{A}_2, \boldsymbol{S}_2)$ are isomorphic. This case frequently occurs in (bio)chemical modelling, where molecules occur in different conformations, but with the same graph topology given by the covalent bonding structure. Recall that each iteration of GWL aggregates geometric information $\boldsymbol{g}_i^{(k)}$ from progressively larger neighbourhoods $\mathcal{N}_i^{(k)}$ around the node $i$, and distinguishes (sub-)graphs via comparing $\mathfrak{G}$-orbit injective colouring of $\boldsymbol{g}_i^{(k)}$. We say $\mathcal{G}_1$ and $\mathcal{G}_2$ are *k-hop distinct* if for all graph isomorphisms $b$, there is some node $i \in \mathcal{V}_1, b(i) \in \mathcal{V}_2$ such that the corresponding $k$-hop subgraphs $\mathcal{N}_i^{(k)}$ and $\mathcal{N}_{b(i)}^{(k)}$ are distinct. Otherwise, we say $\mathcal{G}_1$ and $\mathcal{G}_2$ are *k-hop identical* if all $\mathcal{N}_i^{(k)}$ and $\mathcal{N}_{b(i)}^{(k)}$ are identical up to group actions. We can now formalise what geometric graphs can and cannot be distinguished by GWL.

**Proposition 1** *GWL can distinguish any k-hop distinct geometric graphs $\mathcal{G}_1$ and $\mathcal{G}_2$ where the underlying attributed graphs are isomorphic, and k iterations are sufficient.*

**Proposition 2** *Up to k iterations, GWL cannot distinguish any k-hop identical geometric graphs $\mathcal{G}_1$ and $\mathcal{G}_2$ where the underlying attributed graphs are isomorphic.*

Additionally, we can state the following results about the more constrained IGWL.

**Proposition 3** *IGWL can distinguish any 1-hop distinct geometric graphs $\mathcal{G}_1$ and $\mathcal{G}_2$ where the underlying attributed graphs are isomorphic, and 1 iteration is sufficient.*

**Proposition 4** *Any number of iterations of IGWL cannot distinguish any 1-hop identical geometric graphs $\mathcal{G}_1$ and $\mathcal{G}_2$ where the underlying attributed graphs are isomorphic.*

# Extended Abstract Track

We can now consider the more general case where the underlying attributed graphs for $\mathcal{G}_1 = (\boldsymbol{A}_1, \boldsymbol{S}_1, \overrightarrow{\boldsymbol{V}}_1, \overrightarrow{\boldsymbol{X}}_1)$ and $\mathcal{G}_2 = (\boldsymbol{A}_2, \boldsymbol{S}_2, \overrightarrow{\boldsymbol{V}}_2, \overrightarrow{\boldsymbol{X}}_2)$ are non-isomorphic and constructed from point clouds using radial cutoffs, as conventional in biochemistry and material science.

**Proposition 5** *Assuming geometric graphs are constructed from point clouds using radial cutoffs, GWL can distinguish any geometric graphs $\mathcal{G}_1$ and $\mathcal{G}_2$ where the underlying attributed graphs are non-isomorphic. At most $k_{Max}$ iterations are sufficient, where $k_{Max}$ is the maximum graph diameter among $\mathcal{G}_1$ and $\mathcal{G}_2$.*

These results enable us to compare the expressive powers of GWL and IGWL.

**Theorem 6** *GWL is strictly more powerful than IGWL.*

This statement formalises the advantage of $\mathfrak{G}$-equivariant intermediate layers for graphs and geometric data, as prescribed in the Geometric Deep Learning blueprint (Bronstein et al., 2021), in addition to echoing similar intuitions in the computer vision community. As remarked by (Hinton et al., 2011), translation invariant models do not understand the relationship between the various parts of an image (colloquially called the "Picasso problem"). Similarly, our results explain how IGWL fails to understand how the various 1-hops of a graph are stitched together. Finally, we identify a setting where this distinction between the two approaches disappears.

**Proposition 7** *IGWL has the same expressive power as GWL for fully connected geometric graphs.*

## 2.2. Characterising the Expressive Power of Geometric GNNs

We would like to characterise the maximum expressive power of geometric GNNs based on the GWL test. Firstly, we show that any message passing $\mathfrak{G}$-equivariant GNN can be at most as powerful as GWL in distinguishing non-isomorphic geometric (sub-)graphs.

**Theorem 8** *Any pair of geometric graphs distinguishable by a $\mathfrak{G}$-equivariant GNN is also distinguishable by GWL.*

With sufficient iterations, the output of $\mathfrak{G}$-equivariant GNNs can be equivalent to GWL if certain conditions are met regarding the aggregate, update and readout functions.

**Proposition 9** *$\mathfrak{G}$-equivariant GNNs have the same expressive power as GWL if the following conditions hold: (1) The aggregation $\mathrm{AGG}$ is an injective, $\mathfrak{G}$-equivariant multiset function. (2) The scalar part of the update $\mathrm{UPD}_s$ is a $\mathfrak{G}$-orbit injective, $\mathfrak{G}$-invariant multiset function. (3) The vector part of the update $\mathrm{UPD}_v$ is an injective, $\mathfrak{G}$-equivariant multiset function. (4) The graph-level readout $f$ is an injective multiset function.*

Similar statements can be made for $\mathfrak{G}$-invariant GNNs and IGWL. Thus, we can directly transfer our results about GWL and IGWL to the class of GNNs bounded by the respective tests. This has several interesting practical implications, discussed in Appendix B.

## 3. Conclusion

This work proposes a geometric version of the Weisfeiler-Leman graph isomorphism test (GWL) for discriminating geometric graphs while respecting the underlying spatial symmetries. We use GWL to characterise the expressive power of geometric Graph Neural Networks and demonstrate the advantages of equivariant GNNs over invariant GNNs.

## Acknowledgements

We would like to thank Andrew Blake, David Kovacs, Erik Thiede, Gabor Csanyi, Hannes Stärk, Ilyes Batatia, Iulia Duta, Mario Geiger, Petar Veličković, Soledad Villar, Weihua Hu, and the anonymous reviewers for helpful comments and discussions. CKJ was supported by the A*STAR Singapore National Science Scholarship (PhD). SVM was supported by the UKRI Centre for Doctoral Training in Application of Artificial Intelligence to the study of Environmental Risks (EP/S022961/1).

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

## Appendix A. Background

### A.1. Graph Isomorphism and the Weisfeiler-Leman Test

An attributed graph $\mathcal{G} = (\boldsymbol{A}, \boldsymbol{S})$ with a node set $\mathcal{V}$ of size $n$ consists of an $n \times n$ adjacency matrix $\boldsymbol{A}$ and a matrix of scalar features $\boldsymbol{S} \in \mathbb{R}^{n \times f}$. Two attributed graphs $\mathcal{G}, \mathcal{H}$ are *isomorphic* if there exists an edge-preserving bijection $b : \mathcal{V}(\mathcal{G}) \to \mathcal{V}(\mathcal{H})$ such that $\boldsymbol{s}_i^{(\mathcal{G})} = \boldsymbol{s}_{b(i)}^{(\mathcal{H})}$, where the subscripts index rows and columns in the corresponding matrices.

The *Weisfeiler-Leman test* (WL) is an algorithm for testing whether two (attributed) graphs are isomorphic (Read and Corneil, 1977; Weisfeiler and Leman, 1968). At iteration zero the algorithm assigns a *colour* $c_i^{(0)} \in C$ from a countable space of colours $C$ to each node $i$. Nodes are coloured the same if their features are the same, otherwise, they are coloured differently. In subsequent iterations $t$, WL iteratively updates the node colouring by producing a new $\boldsymbol{c}_i^{(t)} \in C$:

$$\boldsymbol{c}_i^{(t)} := \text{HASH}\left(\boldsymbol{c}_i^{(t-1)}, \{\!\!\{\boldsymbol{c}_j^{(t-1)} \mid j \in \mathcal{N}_i\}\!\!\}\right), \tag{5}$$

where HASH is an injective map (i.e. a perfect hash map) that assigns a unique colour to each input and $\{\!\{\cdot\}\!\}$ denotes a multiset – a set that allows for repeated elements. The test terminates when the partition of the nodes induced by the colours becomes stable. Given two graphs $\mathcal{G}$ and $\mathcal{H}$, if there exists some iteration $t$ for which $\{\!\{c_i^{(t)} \mid i \in \mathcal{V}(\mathcal{G})\}\!\} \neq \{\!\{c_i^{(t)} \mid i \in \mathcal{V}(\mathcal{H})\}\!\}$, then the graphs are not isomorphic. Otherwise, the WL test is inconclusive, and we say it cannot distinguish the two graphs.

Xu et al. (2019); Morris et al. (2019) showed that GNNs are at most as powerful as WL at distinguishing non-isomorphic graphs. Since then, the WL hierarchy has became a powerful tool for analysing the expressive power of GNNs and guided the search for more expressive models (Chen et al., 2019; Maron et al., 2019; Morris et al., 2020; Dwivedi et al., 2020; Bodnar et al., 2021b,a).

### A.2. Group Theory

We assume basic familiarity with group theory, see Zee (2016) for an overview. We denote the action of the group $\mathfrak{G}$ on a space $X$ by $\mathfrak{g} \cdot x$. If $\mathfrak{G}$ acts on spaces $X$ and $Y$, we say a function $f : X \to Y$ is $\mathfrak{G}$-*equivariant* if $f(\mathfrak{g} \cdot x) = \mathfrak{g} \cdot f(x)$. A function $f : X \to Y$ is $\mathfrak{G}$-*invariant* if $f(\mathfrak{g} \cdot x) = f(x)$. The $\mathfrak{G}$-*orbit* of $x \in X$ is $\mathcal{O}_{\mathfrak{G}}(x) = \{\mathfrak{g} \cdot x \mid \mathfrak{g} \in \mathfrak{G}\} \subseteq X$. When $x$ and $x'$ are part of the same orbit, we write $x \simeq x'$. We say a function $f : X \to Y$ is $\mathfrak{G}$-*orbit injective* if we have $f(x_1) = f(x_2)$ if and only if $x_1 \simeq x_2$ for any $x_1, x_2 \in X$. Necessarily, such a function is $\mathfrak{G}$-invariant, since $f(\mathfrak{g} \cdot x) = f(x)$.

We work with the permutation group over $n$ elements $S_n$ and the Lie groups $\mathfrak{G} = SO(d)$ or $\mathfrak{G} = O(d)$. Invariance to the translation group $T(d)$ is conventionally handled using relative positions. Given one of the standard groups above, for an element $\mathfrak{g}$ we denote by $\boldsymbol{M}_{\mathfrak{g}}$ (or another capital letter) its standard matrix representation.

### A.3. Geometric Graphs

Systems in biochemistry (Jamasb et al., 2022), material science (Chanussot et al., 2021), physical simulations (Sanchez-Gonzalez et al., 2020), and multiagent robotics (Li et al., 2020) are conventionally modelled as geometric graphs. For example, molecules are represented as a set of nodes corresponding to atoms, which contain information about the atom type as well as its 3D spatial coordinates and other geometric quantities such as velocity or acceleration. The geometric attributes transform along with Euclidean transformations of the system.

A geometric graph $\mathcal{G} = (\boldsymbol{A}, \boldsymbol{S}, \overrightarrow{\boldsymbol{V}}, \overrightarrow{\boldsymbol{X}})$ with a node set $\mathcal{V}$ is an attributed graph that is also decorated with geometric attributes: node coordinates $\overrightarrow{\boldsymbol{X}} \in \mathbb{R}^{n \times d}$ and (optionally) vector features [1] $\overrightarrow{\boldsymbol{V}} \in \mathbb{R}^{n \times d}$ (*e.g.* velocity, acceleration). The geometric attributes transform as follows under the action of the relevant groups: (1) $S_n$ acts on the graph via $\boldsymbol{P}_{\sigma}\mathcal{G} := (\boldsymbol{P}_{\sigma}\boldsymbol{A}\boldsymbol{P}_{\sigma}^{\top}, \boldsymbol{P}_{\sigma}\boldsymbol{S}, \boldsymbol{P}_{\sigma}\overrightarrow{\boldsymbol{V}}, \boldsymbol{P}_{\sigma}\overrightarrow{\boldsymbol{X}})$; (2) Orthogonal transformations $\boldsymbol{Q}_{\mathfrak{g}} \in \mathfrak{G}$ act on $\overrightarrow{\boldsymbol{V}}, \overrightarrow{\boldsymbol{X}}$ via $\overrightarrow{\boldsymbol{V}}\boldsymbol{Q}_{\mathfrak{g}}, \overrightarrow{\boldsymbol{X}}\boldsymbol{Q}_{\mathfrak{g}}$; and (3) Translations $\overrightarrow{\boldsymbol{t}} \in T(d)$ act on the coordinates $\overrightarrow{\boldsymbol{X}}$ via $\overrightarrow{\boldsymbol{x}}_i + \overrightarrow{\boldsymbol{t}}$ for all nodes $i$. In biochemistry and material science, the conventional procedure for constructing the geometric graph $\mathcal{G} = (\boldsymbol{A}, \boldsymbol{S}, \overrightarrow{\boldsymbol{V}}, \overrightarrow{\boldsymbol{X}})$ is via the underlying point cloud $(\boldsymbol{S}, \overrightarrow{\boldsymbol{V}}, \overrightarrow{\boldsymbol{X}})$ using

---

1. Without loss of generality, we work with a single vector feature per node. Our results generalise to multiple vector features or higher-order geometric tensors per node.

a predetermined radial cutoff $r$. Thus, the adjacency matrix is defined as $a_{ij} = 1$ if $\|\overrightarrow{\boldsymbol{x}}_i - \overrightarrow{\boldsymbol{x}}_j\|_2 \leq r$, or 0 otherwise, for all $a_{ij} \in \boldsymbol{A}$.

Two geometric graphs $\mathcal{G}$ and $\mathcal{H}$ are *geometrically isomorphic* (denoted $\mathcal{G} \simeq \mathcal{H}$) if there exists an attributed graph isomorphism $b$ such that the geometric attributes are equivalent, up to global group actions $\boldsymbol{Q}_{\mathfrak{g}} \in \mathfrak{G}$ and $\overrightarrow{\boldsymbol{t}} \in T(d)$:

$$\left(\boldsymbol{s}_i^{(\mathcal{G})}, \overrightarrow{\boldsymbol{v}}_i^{(\mathcal{G})}, \overrightarrow{\boldsymbol{x}}_i^{(\mathcal{G})}\right) = \left(\boldsymbol{s}_{b(i)}^{(\mathcal{H})}, \boldsymbol{Q}_{\mathfrak{g}} \overrightarrow{\boldsymbol{v}}_{b(i)}^{(\mathcal{H})}, \boldsymbol{Q}_{\mathfrak{g}}(\overrightarrow{\boldsymbol{x}}_{b(i)}^{(\mathcal{H})} + \overrightarrow{\boldsymbol{t}})\right) \quad \text{for all } i \in \mathcal{V}(\mathcal{G}). \tag{6}$$

Geometric graph isomorphism and distinguishing (sub-)graph geometries has important practical implications for representation learning. For *e.g.*, in molecular systems, an ideal architecture should map distinct local structural environments around atoms to distinct embeddings in representation space (Bartók et al., 2013; Pozdnyakov et al., 2020).

### A.4. Geometric Graph Neural Networks

We consider two broad classes of geometric GNN architectures. $\mathfrak{G}$-equivariant GNN layers (Thomas et al., 2018; Anderson et al., 2019; Jing et al., 2020; Satorras et al., 2021; Brandstetter et al., 2022) update scalar and vector features from iteration $t$ to $t+1$ via learnable aggregate and update functions, AGG and UPD, respectively:

$$\boldsymbol{m}_i^{(t)}, \overrightarrow{\boldsymbol{m}}_i^{(t)} := \text{AGG}\left(\{\!\{(\boldsymbol{s}_i^{(t)}, \boldsymbol{s}_j^{(t)}, \overrightarrow{\boldsymbol{v}}_i^{(t)}, \overrightarrow{\boldsymbol{v}}_j^{(t)}, \overrightarrow{\boldsymbol{x}}_{ij}) \mid j \in \mathcal{N}_i\}\!\}\right) \qquad \text{(Aggregate)} \tag{7}$$

$$\boldsymbol{s}_i^{(t+1)}, \overrightarrow{\boldsymbol{v}}_i^{(t+1)} := \text{UPD}\left((\boldsymbol{s}_i^{(t)}, \overrightarrow{\boldsymbol{v}}_i^{(t)}), (\boldsymbol{m}_i^{(t)}, \overrightarrow{\boldsymbol{m}}_i^{(t)})\right) \qquad\qquad \text{(Update)} \tag{8}$$

For *e.g.*, PaiNN (Schütt et al., 2021) interaction layers aggregate scalar and vector features via learnt radial filters:

$$\boldsymbol{s}_i^{(t+1)} := \boldsymbol{s}_i^{(t)} + \sum_{j \in \mathcal{N}_i} f_1\left(\boldsymbol{s}_j^{(t)}, \|\overrightarrow{\boldsymbol{x}}_{ij}\|\right) \tag{9}$$

$$\overrightarrow{\boldsymbol{v}}_i^{(t+1)} := \overrightarrow{\boldsymbol{v}}_i^{(t)} + \sum_{j \in \mathcal{N}_i} f_2\left(\boldsymbol{s}_j^{(t)}, \|\overrightarrow{\boldsymbol{x}}_{ij}\|\right) \odot \overrightarrow{\boldsymbol{v}}_j^{(t)} + \sum_{j \in \mathcal{N}_i} f_3\left(\boldsymbol{s}_j^{(t)}, \|\overrightarrow{\boldsymbol{x}}_{ij}\|\right) \odot \overrightarrow{\boldsymbol{x}}_{ij} \tag{10}$$

Alternatively, $\mathfrak{G}$-invariant layers (Schütt et al., 2018; Xie and Grossman, 2018; Gasteiger et al., 2020) do not update vector features and only aggregate scalar quantities from local neighbourhoods:

$$\boldsymbol{s}_i^{(t+1)} := \text{UPD}\left(\boldsymbol{s}_i^{(t)}, \text{AGG}\left(\{\!\{(\boldsymbol{s}_i^{(t)}, \boldsymbol{s}_j^{(t)}, \overrightarrow{\boldsymbol{v}}_i, \overrightarrow{\boldsymbol{v}}_j, \overrightarrow{\boldsymbol{x}}_{ij}) \mid j \in \mathcal{N}_i\}\!\}\right)\right). \tag{11}$$

For *e.g.*, SchNet (Schütt et al., 2018) uses relative distances to scalarise local geometric information, while DimeNet (Gasteiger et al., 2020) uses both distances and angles, as follows:

$$\boldsymbol{s}_i^{(t+1)} := \boldsymbol{s}_i^{(t)} + \sum_{j \in \mathcal{N}_i} f_1\left(\boldsymbol{s}_j^{(t)}, \|\overrightarrow{\boldsymbol{x}}_{ij}\|\right) \qquad\qquad \text{(SchNet)} \tag{12}$$

$$\boldsymbol{s}_i^{(t+1)} := \sum_{j \in \mathcal{N}_i} f_1\left(\boldsymbol{s}_i^{(t)}, \boldsymbol{s}_j^{(t)}, \sum_{k \in \mathcal{N}_i \setminus \{j\}} f_2\left(\boldsymbol{s}_j^{(t)}, \boldsymbol{s}_k^{(t)}, \|\overrightarrow{\boldsymbol{x}}_{ij}\|, \overrightarrow{\boldsymbol{x}}_{ij} \cdot \overrightarrow{\boldsymbol{x}}_{ik}\right)\right) \quad \text{(DimeNet)} \tag{13}$$

# Extended Abstract Track

For both $\mathfrak{G}$-invariant and $\mathfrak{G}$-equivariant architectures, the scalar features $\{\boldsymbol{s}_i^{(T)}\}$ at the final iteration $T$ are mapped to graph-level features via a permutation-invariant readout $f : \mathbb{R}^{n \times f} \to \mathbb{R}^{f'}$.

Invariant GNNs have shown strong performance for protein design (Zhang et al., 2022; Dauparas et al., 2022) and electrocatalysis (Gasteiger et al., 2021; Shi et al., 2022), while equivariant GNNs are being used within learnt interatomic potentials for molecular dynamics (Schütt et al., 2021; Batzner et al., 2022; Batatia et al., 2022).

## Appendix B. Discussion

**Practical Implications.** Together, Propositions 1 and 4 highlight critical theoretical limitations of $\mathfrak{G}$-invariant GNNs in computing global and non-local geometric properties. Our results suggest that $\mathfrak{G}$-equivariant GNNs should be preferred when working with large geometric graphs such as macromolecules with thousands of nodes, where message passing is restricted to local radial neighbourhoods around each node.

Motivated by these limitations, two straightforward approaches to improving $\mathfrak{G}$-invariant GNNs may be: (1) pre-computing non-local geometric properties as input features, *e.g.* models such as GemNet (Gasteiger et al., 2021) and GearNet (Zhang et al., 2022) successfully use two-hop dihedral angles. And (2) working with fully connected geometric graphs, as Proposition 7 suggests that $\mathfrak{G}$-equivariant and $\mathfrak{G}$-invariant GNNs can be made equally powerful when performing all-to-all message passing. This is supported by the empirical success of recent $\mathfrak{G}$-invariant 'Graph Transformers' (Joshi, 2020; Shi et al., 2022) for small molecules with tens of nodes, where working with full graphs is tractable.

**Related Work.** Literature on the *completeness* of atom-centred interatomic potentials has focused on distinguishing 1-hop local neighbourhoods (point clouds) around atoms by building spanning sets for continuous, $\mathfrak{G}$-equivariant multiset functions (Shapeev, 2016; Drautz, 2019; Dusson et al., 2019; Pozdnyakov et al., 2020). Recent theoretical work on geometric GNNs and their universality has shown that Tensor Field Networks, GemNet and GVP (Dym and Maron, 2020; Gasteiger et al., 2021; Jing et al., 2020; Villar et al., 2021) can be universal approximators of continuous, $\mathfrak{G}$-equivariant or $\mathfrak{G}$-invariant multiset function over point clouds (not sparse graphs). In contrast, the GWL framework studies the expressive power of geometric GNNs from the perspective of geometric graph isomorphism. Overall, our work formalises what classes of geometric graphs can and cannot be distinguished by message passing $\mathfrak{G}$-invariant/equivariant GNNs while abstracting away implementation details.

**Future Work.** GWL provides an abstraction to study the limits of geometric GNNs, but in practice it is challenging to build maximally powerful GNNs that satisfy the conditions of Proposition 9 as GWL relies on $\mathfrak{G}$-orbit injective colouring and $\mathfrak{G}$-equivariant propagation of auxiliary geometric information. Based on the intuitions gained from GWL, future work will explore building provably powerful, *practical* geometric GNNs for applications in biochemistry, material science, and multiagent robotics, and better characterise the trade-offs related to practical implementation choices.

*Extended Abstract Track*

## Appendix C. Proofs for What GWL and IGWL can Distinguish

The following results are a consequence of the construction of GWL as well as the definitions of $k$-hop distinct and $k$-hop identical geometric graphs. Note that $k$-hop distinct geometric graphs are also $(k + 1)$-hop distinct. Similarly, $k$-hop identical geometric graphs are also $(k - 1)$-hop identical, but not necessarily $(k + 1)$-hop distinct.

Given two distinct neighbourhoods $\mathcal{N}_1$ and $\mathcal{N}_2$, the $\mathfrak{G}$-orbits of the corresponding geometric multisets $\boldsymbol{g}_1$ and $\boldsymbol{g}_2$ are mutually exclusive, *i.e.* $\mathcal{O}_{\mathfrak{G}}(\boldsymbol{g}_1) \cap \mathcal{O}_{\mathfrak{G}}(\boldsymbol{g}_2) \equiv \emptyset$. By the properties of I-HASH this implies $c_1 \neq c_2$. Conversely, if $\mathcal{N}_1$ and $\mathcal{N}_2$ were identical up to group actions, their $\mathfrak{G}$-orbits would overlap, *i.e.* $\boldsymbol{g}_1 = \mathfrak{g} \, \boldsymbol{g}_2$ for some $\mathfrak{g} \in \mathfrak{G}$ and $\mathcal{O}_{\mathfrak{G}}(\boldsymbol{g}_1) = \mathcal{O}_{\mathfrak{G}}(\boldsymbol{g}_2) \Rightarrow c_1 = c_2$.

**Proposition 10** *GWL can distinguish any $k$-hop distinct geometric graphs $\mathcal{G}_1$ and $\mathcal{G}_2$ where the underlying attributed graphs are isomorphic, and $k$ iterations are sufficient.*

**Proof** [Proof of Proposition 1] The $k$-th iteration of GWL identifies the $\mathfrak{G}$-orbit of the $k$-hop subgraph $\mathcal{N}_i^{(k)}$ at each node $i$ via the geometric multiset $\boldsymbol{g}_i^{(k)}$. $\mathcal{G}_1$ and $\mathcal{G}_2$ being $k$-hop distinct implies that there exists some bijection $b$ and some node $i \in \mathcal{V}_1, b(i) \in \mathcal{V}_2$ such that the corresponding $k$-hop subgraphs $\mathcal{N}_i^{(k)}$ and $\mathcal{N}_{b(i)}^{(k)}$ are distinct. Thus, the $\mathfrak{G}$-orbits of the corresponding geometric multisets $\boldsymbol{g}_i^{(k)}$ and $\boldsymbol{g}_{b(i)}^{(k)}$ are mutually exclusive, *i.e.* $\mathcal{O}_{\mathfrak{G}}(\boldsymbol{g}_i^{(k)}) \cap \mathcal{O}_{\mathfrak{G}}(\boldsymbol{g}_{b(i)}^{(k)}) \equiv \emptyset \Rightarrow c_i^{(k)} \neq c_{b(i)}^{(k)}$. Thus, $k$ iterations of GWL are sufficient to distinguish $\mathcal{G}_1$ and $\mathcal{G}_2$. ∎

**Proposition 11** *Up to $k$ iterations, GWL cannot distinguish any $k$-hop identical geometric graphs $\mathcal{G}_1$ and $\mathcal{G}_2$ where the underlying attributed graphs are isomorphic.*

**Proof** [Proof of Proposition 2] The $k$-th iteration of GWL identifies the $\mathfrak{G}$-orbit of the $k$-hop subgraph $\mathcal{N}_i^{(k)}$ at each node $i$ via the geometric multiset $\boldsymbol{g}_i^{(k)}$. $\mathcal{G}_1$ and $\mathcal{G}_2$ being $k$-hop identical implies that for all bijections $b$ and all nodes $i \in \mathcal{V}_1, b(i) \in \mathcal{V}_2$, the corresponding $k$-hop subgraphs $\mathcal{N}_i^{(k)}$ and $\mathcal{N}_{b(i)}^{(k)}$ are identical up to group actions. Thus, the $\mathfrak{G}$-orbits of the corresponding geometric multisets $\boldsymbol{g}_i^{(k)}$ and $\boldsymbol{g}_{b(i)}^{(k)}$ overlap, *i.e.* $\mathcal{O}_{\mathfrak{G}}(\boldsymbol{g}_i^{(k)}) = \mathcal{O}_{\mathfrak{G}}(\boldsymbol{g}_{b(i)}^{(k)}) \Rightarrow c_i^{(k)} = c_{b(i)}^{(k)}$. Thus, up to $k$ iterations of GWL cannot distinguish $\mathcal{G}_1$ and $\mathcal{G}_2$. ∎

**Proposition 12** *IGWL can distinguish any 1-hop distinct geometric graphs $\mathcal{G}_1$ and $\mathcal{G}_2$ where the underlying attributed graphs are isomorphic, and 1 iteration is sufficient.*

**Proof** [Proof of Proposition 3] Each iteration of IGWL identifies the $\mathfrak{G}$-orbit of the 1-hop local neighbourhood $\mathcal{N}_i^{(k=1)}$ at each node $i$. $\mathcal{G}_1$ and $\mathcal{G}_2$ being 1-hop distinct implies that there exists some bijection $b$ and some node $i \in \mathcal{V}_1, b(i) \in \mathcal{V}_2$ such that the corresponding 1-hop local neighbourhoods $\mathcal{N}_i^{(1)}$ and $\mathcal{N}_{b(i)}^{(1)}$ are distinct. Thus, the $\mathfrak{G}$-orbits of the corresponding geometric multisets $\boldsymbol{g}_i^{(1)}$ and $\boldsymbol{g}_{b(i)}^{(1)}$ are mutually exclusive, *i.e.* $\mathcal{O}_{\mathfrak{G}}(\boldsymbol{g}_i^{(1)}) \cap \mathcal{O}_{\mathfrak{G}}(\boldsymbol{g}_{b(i)}^{(1)}) \equiv \emptyset \Rightarrow c_i^{(1)} \neq c_{b(i)}^{(1)}$. Thus, 1 iteration of IGWL is sufficient to distinguish $\mathcal{G}_1$ and $\mathcal{G}_2$. ∎

Extended Abstract Track

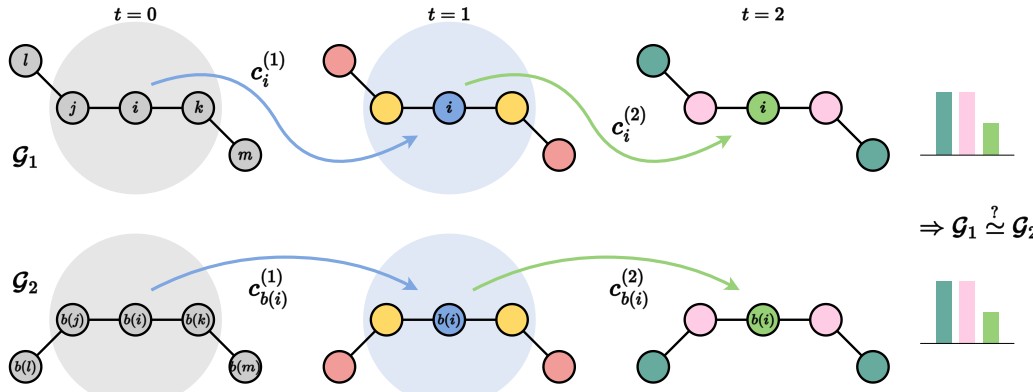

Figure 2: **Invariant GWL Test.** IGWL cannot distinguish $\mathcal{G}_1$ and $\mathcal{G}_2$ as they are 1-hop identical: The $\mathfrak{G}$-orbit of the 1-hop neighbourhood around each node is the same, and IGWL cannot propagate geometric orientation information beyond 1-hop (here $\mathfrak{G} = O(d)$).

**Proposition 13** *Any number of iterations of IGWL cannot distinguish any 1-hop identical geometric graphs $\mathcal{G}_1$ and $\mathcal{G}_2$ where the underlying attributed graphs are isomorphic.*

**Proof** [Proof of Proposition 4] Each iteration of IGWL identifies the $\mathfrak{G}$-orbit of the 1-hop local neighbourhood $\mathcal{N}_i^{(k=1)}$ at each node $i$, but cannot identify $\mathfrak{G}$-orbits beyond 1-hop by the construction of IGWL as no geometric information is propagated. $\mathcal{G}_1$ and $\mathcal{G}_2$ being 1-hop identical implies that for all bijections $b$ and all nodes $i \in \mathcal{V}_1, b(i) \in \mathcal{V}_2$, the corresponding 1-hop local neighbourhoods $\mathcal{N}_i^{(k)}$ and $\mathcal{N}_{b(i)}^{(k)}$ are identical up to group actions. Thus, the $\mathfrak{G}$-orbits of the corresponding geometric multisets $\boldsymbol{g}_i^{(1)}$ and $\boldsymbol{g}_{b(i)}^{(1)}$ overlap, *i.e.* $\mathcal{O}_{\mathfrak{G}}(\boldsymbol{g}_i^{(1)}) = \mathcal{O}_{\mathfrak{G}}(\boldsymbol{g}_{b(i)}^{(1)}) \Rightarrow c_i^{(k)} = c_{b(i)}^{(k)}$. Thus, any number of IGWL iterations cannot distinguish $\mathcal{G}_1$ and $\mathcal{G}_2$. ∎

**Proposition 14** *Assuming geometric graphs are constructed from point clouds using radial cutoffs, GWL can distinguish any geometric graphs $\mathcal{G}_1$ and $\mathcal{G}_2$ where the underlying attributed graphs are non-isomorphic. At most $k_{Max}$ iterations are sufficient, where $k_{Max}$ is the maximum graph diameter among $\mathcal{G}_1$ and $\mathcal{G}_2$.*

**Proof** [Proof of Proposition 5] We assume that a geometric graph $\mathcal{G} = (\boldsymbol{A}, \boldsymbol{S}, \vec{\boldsymbol{V}}, \vec{\boldsymbol{X}})$ is constructed from a point cloud $(\boldsymbol{S}, \vec{\boldsymbol{V}}, \vec{\boldsymbol{X}})$ using a predetermined radial cutoff $r$. Thus, the adjacency matrix is defined as $a_{ij} = 1$ if $\|\vec{\boldsymbol{x}}_i - \vec{\boldsymbol{x}}_j\|_2 \leq r$, or 0 otherwise, for all $a_{ij} \in \boldsymbol{A}$. Such construction procedures are conventional for geometric graphs in biochemistry and material science.

Given geometric graphs $\mathcal{G}_1$ and $\mathcal{G}_2$ where the underlying attributed graphs are non-isomorphic, identify $k_{\text{Max}}$ the maximum of the graph diameters of $\mathcal{G}_1$ and $\mathcal{G}_2$, and chose any arbitrary nodes $i \in \mathcal{V}_1, j \in \mathcal{V}_2$. We can define the $k_{\text{Max}}$-hop subgraphs $\mathcal{N}_i^{(k_{\text{Max}})}$ and $\mathcal{N}_j^{(k_{\text{Max}})}$

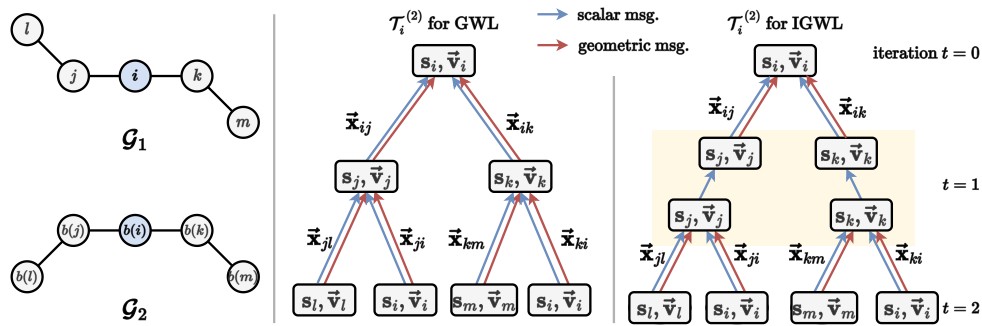

Figure 3: **Geometric Computation Trees for GWL and IGWL.** A computation tree $\mathcal{T}_i^{(t)}$ represents the maximum information contained in GWL/IGWL colours or GNN features at iteration $t$ by an 'unrolling' of the message passing procedure. Unlike GWL, geometric orientation information cannot flow from the leaves to the root in IGWL, restricting its expressive power. IGWL cannot distinguish $\mathcal{G}_1$ and $\mathcal{G}_2$ as all 1-hop neighbourhoods are computationally identical.

at $i$ and $j$, respectively. Thus, $\mathcal{N}_i^{(k_{\mathrm{Max}})} = \mathcal{V}_1$ for all $i \in \mathcal{V}_1$, and $\mathcal{N}_j^{(k_{\mathrm{Max}})} = \mathcal{V}_2$ for all $j \in \mathcal{V}_2$. Due to the assumed construction procedure of geometric graphs, $\mathcal{N}_i^{(k_{\mathrm{Max}})}$ and $\mathcal{N}_j^{(k_{\mathrm{Max}})}$ must be distinct. Otherwise, if $\mathcal{N}_i^{(k_{\mathrm{Max}})}$ and $\mathcal{N}_j^{(k_{\mathrm{Max}})}$ were identical up to group actions, the sets $(\boldsymbol{S}_1, \overrightarrow{\boldsymbol{V}}_1, \overrightarrow{\boldsymbol{X}}_1)$ and $(\boldsymbol{S}_2, \overrightarrow{\boldsymbol{V}}_2, \overrightarrow{\boldsymbol{X}}_2)$ would have yielded isomorphic graphs.

The $k_{\mathrm{Max}}$-th iteration of GWL identifies the $\mathfrak{G}$-orbit of the $k_{\mathrm{Max}}$-hop subgraph $\mathcal{N}_i^{(k_{\mathrm{Max}})}$ at each node $i$ via the geometric multiset $\boldsymbol{g}_i^{(k_{\mathrm{Max}})}$. As $\mathcal{N}_i^{(k_{\mathrm{Max}})}$ and $\mathcal{N}_j^{(k_{\mathrm{Max}})}$ are distinct for any arbitrary nodes $i \in \mathcal{V}_1, j \in \mathcal{V}_2$, the $\mathfrak{G}$-orbits of the corresponding geometric multisets $\boldsymbol{g}_i^{(k_{\mathrm{Max}})}$ and $\boldsymbol{g}_j^{(k_{\mathrm{Max}})}$ are mutually exclusive, *i.e.* $\mathcal{O}_{\mathfrak{G}}(\boldsymbol{g}_i^{(k_{\mathrm{Max}})}) \cap \mathcal{O}_{\mathfrak{G}}(\boldsymbol{g}_j^{(k_{\mathrm{Max}})}) \equiv \emptyset \Rightarrow c_i^{(k_{\mathrm{Max}})} \neq c_j^{(k_{\mathrm{Max}})}$. Thus, $k_{\mathrm{Max}}$ iterations of GWL are sufficient to distinguish $\mathcal{G}_1$ and $\mathcal{G}_2$. ∎

**Theorem 6** *GWL is strictly more powerful than IGWL.*

**Proof** [Proof of Theorem 6]

Firstly, we can show that the GWL class contains IGWL if GWL can learn the identity when updating $\boldsymbol{g}_i$ for all $i \in \mathcal{V}$, *i.e.* $\boldsymbol{g}_i^{(t)} = \boldsymbol{g}_i^{(t-1)} = \boldsymbol{g}_i^{(0)} \equiv (\boldsymbol{s}_i, \overrightarrow{\boldsymbol{v}}_i)$. Thus, GWL is at least as powerful as IGWL, which does not update $\boldsymbol{g}_i$.

Secondly, to show that GWL is strictly more powerful than IGWL, it suffices to show that there exist a pair of geometric graphs that can be distinguished by GWL but not by IGWL. We may consider any $k$-hop distinct geometric graphs for $k > 1$, where the underlying attributed graphs are isomorphic. Proposition 1 states that GWL can distinguish any such graphs, while Proposition 4 states that IGWL cannot distinguish them. An example is the pair of graphs in Figures 1 and 2. ∎

# Extended Abstract Track

**Proposition 15** *IGWL has the same expressive power as GWL for fully connected geometric graphs.*

**Proof** [Proof of Proposition 7] We will prove by contradiction. Assume that there exist a pair of fully connected geometric graphs $\mathcal{G}_1$ and $\mathcal{G}_2$ which GWL can distinguish, but IGWL cannot.

If the underlying attributed graphs of $\mathcal{G}_1$ and $\mathcal{G}_2$ are isomorphic, by Proposition 1 and Proposition 4, $\mathcal{G}_1$ and $\mathcal{G}_2$ are 1-hop identical but $k$-hop distinct for some $k > 1$. For all bijections $b$ and all nodes $i \in \mathcal{V}_1, b(i) \in \mathcal{V}_2$, the local neighbourhoods $\mathcal{N}_i^{(1)}$ and $\mathcal{N}_{b(i)}^{(1)}$ are identical up to group actions, and $\mathcal{O}_{\mathfrak{G}}(\boldsymbol{g}_i^{(1)}) = \mathcal{O}_{\mathfrak{G}}(\boldsymbol{g}_{b(i)}^{(1)}) \Rightarrow c_i^{(1)} = c_{b(i)}^{(1)}$. Additionally, there exists some bijection $b$ and some nodes $i \in \mathcal{V}_1, b(i) \in \mathcal{V}_2$ such that the $k$-hop subgraphs $\mathcal{N}_i^{(k)}$ and $\mathcal{N}_{b(i)}^{(k)}$ are distinct, and $\mathcal{O}_{\mathfrak{G}}(\boldsymbol{g}_i^{(k)}) \cap \mathcal{O}_{\mathfrak{G}}(\boldsymbol{g}_{b(i)}^{(k)}) \equiv \emptyset \Rightarrow c_i^{(k)} \neq c_{b(i)}^{(k)}$. However, as $\mathcal{G}_1$ and $\mathcal{G}_2$ are fully connected, for any $k$, $\mathcal{N}_i^{(1)} = \mathcal{N}_i^{(k)}$ and $\mathcal{N}_{b(i)}^{(1)} = \mathcal{N}_{b(i)}^{(k)}$ are identical up to group actions. Thus, $\mathcal{O}_{\mathfrak{G}}(\boldsymbol{g}_i^{(1)}) = \mathcal{O}_{\mathfrak{G}}(\boldsymbol{g}_i^{(k)}) = \mathcal{O}_{\mathfrak{G}}(\boldsymbol{g}_{b(i)}^{(1)}) = \mathcal{O}_{\mathfrak{G}}(\boldsymbol{g}_{b(i)}^{(k)}) \Rightarrow c_i^{(1)} = c_i^{(k)} = c_{b(i)}^{(k)} = c_{b(i)}^{(k)}$. This is a contradiction.

If $\mathcal{G}_1$ and $\mathcal{G}_2$ are non-isomorphic and fully connected, for any arbitrary $i \in \mathcal{V}_1, j \in \mathcal{V}_2$ and any $k$-hop neighbourhood, we know that $\mathcal{N}_i^{(1)} = \mathcal{N}_i^{(k)}$ and $\mathcal{N}_j^{(1)} = \mathcal{N}_j^{(k)}$. Thus, a single iteration of GWL and IGWL identify the same $\mathfrak{G}$-orbits and assign the same node colours, *i.e.* $\mathcal{O}_{\mathfrak{G}}(\boldsymbol{g}_i^{(1)}) = \mathcal{O}_{\mathfrak{G}}(\boldsymbol{g}_i^{(k)}) \Rightarrow c_i^{(1)} = c_i^{(k)}$ and $\mathcal{O}_{\mathfrak{G}}(\boldsymbol{g}_j^{(1)}) = \mathcal{O}_{\mathfrak{G}}(\boldsymbol{g}_j^{(k)}) \Rightarrow c_j^{(1)} = c_j^{(k)}$. This is a contradiction. ∎

## Appendix D. Proofs for equivalence between GWL and Geometric GNNs

Our proofs adapt the techniques used in (Xu et al., 2019; Morris et al., 2019) for connecting 1-WL with GNNs. Note that we omit including the relative position vectors $\overrightarrow{\boldsymbol{x}}_{ij} = \overrightarrow{\boldsymbol{x}}_i - \overrightarrow{\boldsymbol{x}}_j$ in GWL and geometric GNN updates for brevity, as relative positions vectors can be merged into the vector features.

**Theorem 8** *Any pair of geometric graphs distinguishable by a $\mathfrak{G}$-equivariant GNN is also distinguishable by GWL.*

**Proof** [**Proof of Theorem 8**]

Consider two geometric graphs $\mathcal{G}$ and $\mathcal{H}$. The theorem implies that if the GNN graph-level readout outputs $f(\mathcal{G}) \neq f(\mathcal{H})$, then the GWL test will always determine $\mathcal{G}$ and $\mathcal{H}$ to be non-isomorphic, *i.e.* $\mathcal{G} \neq \mathcal{H}$.

We will prove by contradiction. Suppose after $T$ iterations, a GNN graph-level readout outputs $f(\mathcal{G}) \neq f(\mathcal{H})$, but the GWL test cannot decide $\mathcal{G}$ and $\mathcal{H}$ are non-isomorphic, *i.e.* $\mathcal{G}$ and $\mathcal{H}$ always have the same collection of node colours for iterations 0 to $T$. Thus, for iteration $t$ and $t + 1$ for any $t = 0 \ldots T - 1$, $\mathcal{G}$ and $\mathcal{H}$ have the same collection of node colours $\{c_i^{(t)}\}$ as well as the same collection of neighbourhood geometric multisets $\left\{ (c_i^{(t)}, \boldsymbol{g}_i^{(t)}) , \{\!\{ (c_j^{(t)}, \boldsymbol{g}_j^{(t)}) \mid j \in \mathcal{N}_i \}\!\} \right\}$ up to group actions. Otherwise, the GWL test would have produced different node colours at iteration $t + 1$ for $\mathcal{G}$ and $\mathcal{H}$ as different geometric multisets get unique new colours.

# Extended Abstract Track

We will show that on the same graph for nodes $i$ and $k$, if $(c_i^{(t)}, \boldsymbol{g}_i^{(t)}) = (c_k^{(t)}, \mathfrak{g} \cdot \boldsymbol{g}_k^{(t)})$, we always have GNN features $(\boldsymbol{s}_i^{(t)}, \overrightarrow{\boldsymbol{v}}_i^{(t)}) = (\boldsymbol{s}_k^{(t)}, \boldsymbol{Q}_\mathfrak{g} \overrightarrow{\boldsymbol{v}}_k^{(t)})$ for any iteration $t$. This holds for $t = 0$ because GWL and the GNN start with the same initialisation. Suppose this holds for iteration $t$. At iteration $t + 1$, if for any $i$ and $k$, $(c_i^{(t+1)}, \boldsymbol{g}_i^{(t+1)}) = (c_k^{(t+1)}, \mathfrak{g} \cdot \boldsymbol{g}_k^{(t+1)})$, then:

$$\left\{ (c_i^{(t)}, \boldsymbol{g}_i^{(t)}) \, , \, \{\!\{ (c_j^{(t)}, \boldsymbol{g}_j^{(t)}) \mid j \in \mathcal{N}_i \}\!\} \right\} = \left\{ (c_k^{(t)}, \mathfrak{g} \cdot \boldsymbol{g}_k^{(t)}) \, , \, \{\!\{ (c_j^{(t)}, \mathfrak{g} \cdot \boldsymbol{g}_j^{(t)}) \mid j \in \mathcal{N}_k \}\!\} \right\} \quad (14)$$

By our assumption on iteration $t$,

$$\left\{ (\boldsymbol{s}_i^{(t)}, \overrightarrow{\boldsymbol{v}}_i^{(t)}) \, , \, \{\!\{ (\boldsymbol{s}_j^{(t)}, \overrightarrow{\boldsymbol{v}}_j^{(t)}) \mid j \in \mathcal{N}_i \}\!\} \right\} = \left\{ (\boldsymbol{s}_k^{(t)}, \boldsymbol{Q}_\mathfrak{g} \overrightarrow{\boldsymbol{v}}_k^{(t)}) \, , \, \{\!\{ (\boldsymbol{s}_j^{(t)}, \boldsymbol{Q}_\mathfrak{g} \overrightarrow{\boldsymbol{v}}_j^{(t)}) \mid j \in \mathcal{N}_k \}\!\} \right\}$$
$$(15)$$

As the same aggregate and update operations are applied at each node within the GNN, the same inputs, *i.e.* neighbourhood features, are mapped to the same output. Thus, $(\boldsymbol{s}_i^{(t+1)}, \overrightarrow{\boldsymbol{v}}_i^{(t+1)}) = (\boldsymbol{s}_k^{(t+1)}, \boldsymbol{Q}_\mathfrak{g} \overrightarrow{\boldsymbol{v}}_k^{(t+1)})$. By induction, if $(c_i^{(t)}, \boldsymbol{g}_i^{(t)}) = (c_k^{(t)}, \mathfrak{g} \cdot \boldsymbol{g}_k^{(t)})$, we always have GNN node features $(\boldsymbol{s}_i^{(t)}, \overrightarrow{\boldsymbol{v}}_i^{(t)}) = (\boldsymbol{s}_k^{(t)}, \boldsymbol{Q}_\mathfrak{g} \overrightarrow{\boldsymbol{v}}_k^{(t)})$ for any iteration $t$. This creates valid mappings $\phi_s, \phi_v$ such that $\boldsymbol{s}_i^{(t)} = \phi_s(c_i^{(t)})$ and $\overrightarrow{\boldsymbol{v}}_i^{(t)} = \phi_v(c_i^{(t)}, \boldsymbol{g}_i^{(t)})$ for any $i \in \mathcal{V}$.

Thus, if $\mathcal{G}$ and $\mathcal{H}$ have the same collection of node colours and geometric multisets, then $\mathcal{G}$ and $\mathcal{H}$ also have the same collection of GNN neighbourhood features

$$\left\{ (\boldsymbol{s}_i^{(t)}, \overrightarrow{\boldsymbol{v}}_i^{(t)}) \, , \, \{\!\{ (\boldsymbol{s}_j^{(t)}, \overrightarrow{\boldsymbol{v}}_j^{(t)}) \mid j \in \mathcal{N}_i \}\!\} \right\} = \left\{ (\phi_s(c_i^{(t)}), \phi_v(c_i^{(t)}, \boldsymbol{g}_i^{(t)})) \, , \, \{\!\{ (\phi_s(c_j^{(t)}), \phi_v(c_i^{(t)}, \boldsymbol{g}_i^{(t)})) \mid j \in \mathcal{N}_i \}\!\} \right\}$$

Thus, the GNN will output the same collection of node scalar features $\{\boldsymbol{s}_i^{(T)}\}$ for $\mathcal{G}$ and $\mathcal{H}$ and the permutation-invariant graph-level readout will output $f(\mathcal{G}) = f(\mathcal{H})$. This is a contradiction. ∎

**Proposition 16** $\mathfrak{G}$-*equivariant GNNs have the same expressive power as GWL if the following conditions hold: (1) The aggregation* AGG *is an injective,* $\mathfrak{G}$-*equivariant multiset function. (2) The scalar part of the update* UPD$_s$ *is a* $\mathfrak{G}$-*orbit injective,* $\mathfrak{G}$-*invariant multiset function. (3) The vector part of the update* UPD$_v$ *is an injective,* $\mathfrak{G}$-*equivariant multiset function. (4) The graph-level readout* $f$ *is an injective multiset function.*

**Proof [Proof of Theorem 9]**

Consider a GNN where the conditions hold. We will show that, with a sufficient number of iterations $t$, the output of this GNN is equivalent to GWL, *i.e.* $\boldsymbol{s}^{(t)} \equiv c^{(t)}$.

Let $\mathcal{G}$ and $\mathcal{H}$ be any geometric graphs which the GWL test decides as non-isomorphic at iteration $T$. Because the graph-level readout function is injective, *i.e.* it maps distinct multiset of node scalar features into unique embeddings, it suffices to show that the GNN's neighbourhood aggregation process, with sufficient iterations, embeds $\mathcal{G}$ and $\mathcal{H}$ into different multisets of node features.

For this proof, we replace $\mathfrak{G}$-orbit injective functions with injective functions over the equivalence class generated by the actions of $\mathfrak{G}$. Thus, all elements belonging to the same $\mathfrak{G}$-orbit will first be mapped to the same representative of the equivalence class, denoted by the square brackets $[\ldots]$, followed by an injective map. The result is $\mathfrak{G}$-orbit injective.

# Extended Abstract Track

Let us assume the GNN updates node scalar and vector features as:

$$\boldsymbol{s}_i^{(t)} = \text{UPD}_s \left( \left[ (\boldsymbol{s}_i^{(t-1)}, \overrightarrow{\boldsymbol{v}}_i^{(t-1)}) \ , \ \text{AGG} \left( \{\!\{ (\boldsymbol{s}_i^{(t-1)}, \boldsymbol{s}_j^{(t-1)}, \overrightarrow{\boldsymbol{v}}_i^{(t-1)}, \overrightarrow{\boldsymbol{v}}_j^{(t-1)}) \mid j \in \mathcal{N}_i \}\!\} \right) \right] \right)$$
(16)

$$\overrightarrow{\boldsymbol{v}}_i^{(t)} = \text{UPD}_v \left( (\boldsymbol{s}_i^{(t-1)}, \overrightarrow{\boldsymbol{v}}_i^{(t-1)}) \ , \ \text{AGG} \left( \{\!\{ (\boldsymbol{s}_i^{(t-1)}, \boldsymbol{s}_j^{(t-1)}, \overrightarrow{\boldsymbol{v}}_i^{(t-1)}, \overrightarrow{\boldsymbol{v}}_j^{(t-1)}) \mid j \in \mathcal{N}_i \}\!\} \right) \right) \quad (17)$$

with the aggregation function AGG being $\mathfrak{G}$-equivariant and injective, the scalar update function $\text{UPD}_s$ being $\mathfrak{G}$-invariant and injective, and the vector update function $\text{UPD}_v$ being $\mathfrak{G}$-equivariant and injective.

The GWL test updates the node colour $c_i^{(t)}$ and geometric multiset $\boldsymbol{g}_i^{(t)}$ as:

$$c_i^{(t)} = h_s \left( \left[ (c_i^{(t-1)}, \boldsymbol{g}_i^{(t-1)}) \ , \ \{\!\{ (c_j^{(t-1)}, \boldsymbol{g}_j^{(t-1)}) \mid j \in \mathcal{N}_i \}\!\} \right] \right),$$
(18)

$$\boldsymbol{g}_i^{(t)} = h_v \left( (c_i^{(t-1)}, \boldsymbol{g}_i^{(t-1)}) \ , \ \{\!\{ (c_j^{(t-1)}, \boldsymbol{g}_j^{(t-1)}) \mid j \in \mathcal{N}_i \}\!\} \right),$$
(19)

where $h_s$ is a $\mathfrak{G}$-invariant and injective map, and $h_v$ is a $\mathfrak{G}$-equivariant and injective operation (e.g. in equation 2, expanding the geometric multiset by copying).

We will show by induction that at any iteration $t$, there always exist injective functions $\varphi_s$ and $\varphi_v$ such that $\boldsymbol{s}_i^{(t)} = \varphi_s(c_i^{(t)})$ and $\overrightarrow{\boldsymbol{v}}_i^{(t)} = \varphi_v(c_i^{(t)}, \boldsymbol{g}_i^{(t)})$. This holds for $t = 0$ because the initial node features are the same for GWL and GNN, $c_i^{(0)} \equiv \boldsymbol{s}_i^{(0)}$ and $\boldsymbol{g}_i^{(0)} \equiv (\boldsymbol{s}_i^{(0)}, \overrightarrow{\boldsymbol{v}}_i^{(0)})$ for all $i \in \mathcal{V}(\mathcal{G}), \mathcal{V}(\mathcal{H})$. Suppose this holds for iteration $t$. At iteration $t + 1$, substituting $\boldsymbol{s}_i^{(t)}$ with $\varphi_s(c_i^{(t)})$, and $\overrightarrow{\boldsymbol{v}}_i^{(t)}$ with $\varphi_v(c_i^{(t)}, \boldsymbol{g}_i^{(t)})$ gives us

$$\boldsymbol{s}_i^{(t+1)} = \text{UPD}_s \left( \left[ (\varphi_s(c_i^{(t)}), \varphi_v(c_i^{(t)}, \boldsymbol{g}_i^{(t)})) \ , \ \text{AGG} \left( \{\!\{ (\varphi_s(c_i^{(t)}), \varphi_s(c_j^{(t)}), \varphi_v(c_i^{(t)}, \boldsymbol{g}_i^{(t)}), \varphi_v(c_j^{(t)}, \boldsymbol{g}_j^{(t)})) \mid j \in \mathcal{N}_i \}\!\} \right) \right] \right)$$

$$\overrightarrow{\boldsymbol{v}}_i^{(t+1)} = \text{UPD}_v \left( (\varphi_s(c_i^{(t)}), \varphi_v(c_i^{(t)}, \boldsymbol{g}_i^{(t)})) \ , \ \text{AGG} \left( \{\!\{ (\varphi_s(c_i^{(t)}), \varphi_s(c_j^{(t)}), \varphi_v(c_i^{(t)}, \boldsymbol{g}_i^{(t)}), \varphi_v(c_j^{(t)}, \boldsymbol{g}_j^{(t)})) \mid j \in \mathcal{N}_i \}\!\} \right) \right)$$

The composition of multiple injective functions is injective. Therefore, there exist some injective functions $g_s$ and $g_v$ such that:

$$\boldsymbol{s}_i^{(t+1)} = g_s \left( \left[ (c_i^{(t)}, \boldsymbol{g}_i^{(t)}) \ , \ \{\!\{ (c_j^{(t)}, \boldsymbol{g}_j^{(t)}) \mid j \in \mathcal{N}_i \}\!\} \right] \right),$$
(20)

$$\overrightarrow{\boldsymbol{v}}_i^{(t+1)} = g_v \left( (c_i^{(t)}, \boldsymbol{g}_i^{(t)}) \ , \ \{\!\{ (c_j^{(t)}, \boldsymbol{g}_j^{(t)}) \mid j \in \mathcal{N}_i \}\!\} \right),$$
(21)

We can then consider:

$$\boldsymbol{s}_i^{(t+1)} = g_s \circ h_s^{-1} \ h_s \left( \left[ (c_i^{(t)}, \boldsymbol{g}_i^{(t)}) \ , \ \{\!\{ (c_j^{(t)}, \boldsymbol{g}_j^{(t)}) \mid j \in \mathcal{N}_i \}\!\} \right] \right),$$
(22)

$$\overrightarrow{\boldsymbol{v}}_i^{(t+1)} = g_v \circ h_v^{-1} \ h_v \left( (c_i^{(t)}, \boldsymbol{g}_i^{(t)}) \ , \ \{\!\{ (c_j^{(t)}, \boldsymbol{g}_j^{(t)}) \mid j \in \mathcal{N}_i \}\!\} \right),$$
(23)

Then, we can denote $\varphi_s = g_s \circ h_s^{-1}$ and $\varphi_v = g_v \circ h_v^{-1}$ as injective functions because the composition of injective functions is injective. Hence, for any iteration $t+1$, there exist injective functions $\varphi_s$ and $\varphi_v$ such that $\boldsymbol{s}_i^{(t+1)} = \varphi_s \left( c_i^{(t+1)} \right)$ and $\overrightarrow{\boldsymbol{v}}_i^{(t+1)} = \varphi_v \left( c_i^{(t+1)}, \boldsymbol{g}_i^{(t+1)} \right)$. At the $T$-th iteration, the GWL test decides that $\mathcal{G}$ and $\mathcal{H}$ are non-isomorphic, which means the multisets of node colours $\{c_i^{(T)}\}$ are different for $\mathcal{G}$ and $\mathcal{H}$. The GNN's node scalar features $\{\boldsymbol{s}_i^{(T)}\} = \{\varphi_s(c_i^{(T)})\}$ must also be different for $\mathcal{G}$ and $\mathcal{H}$ because of the injectivity of $\varphi_s$. ∎

