# OpenReview forum: "On the Expressive Power of Geometric Graph Neural Networks"
_NeurIPS.cc/2022/Workshop/NeurReps — NeurReps 2022 Oral_

### Official Review · Reviewer_6wgv · 2022-10-10
**A theoretical contribution with promising analysis on geometric graphs**

**Confidence:** 3
**Soundness:** 3
**Presentation:** 3
**Contribution:** 3
**Overall Rating:** 6

**Summary:**

The authors propose a new version of the Weisfeiler-Leman test ad-hoc for geometric graphs (**GWL**)  that respect symmetries such as rotation, permutations and translation. They first define GWL and invariant GWL (IGWL) and then demonstrate properties and theorems on them. In particular, they use GWL to understand which geometric graphs can and cannot be distinguished by GNN invariant and equivariant to **spatial symmetries** and show that GWL is more powerful than IGWL.

**Questions:**

The demonstrations of Propositions 3-4 rely on the fact that “The $k$-th iteration of GWL identifies the G-orbit of the $k$-hop subgraph $\mathcal{N}^{(k)}_i$ at each node $i$ via the geometric multiset $\mathbf{g}_i^{(k)}$ ”. Was this demonstrated? Could you provide a reference or clarify the theory behind this statement?

I would ask the authors to clarify the typo on page 3 previously specified.


**Limitations:**

The authors may discuss the computational burden of GWL and IGWL, for instance, comparing their results between them or with the standard WL test.

**Recommended Decision:**

3: Accept

**Relevance:**

4: Highly relevant

**Strengths And Weaknesses:**

The paper adapts the WL test to geometric graphs. I consider the idea quite simple, but the main contribution that the paper can bring is the analysis of geometric graphs. In particular, I think the interesting case is when 2 graphs are isomorphic but different up to rotation or translation.

Sound demonstrations support all the propositions and theorems.

In some parts, the sentences are confusing. At the beginning of page 3, there may be some typos:
- “standard matrix representation of $\mathcal{G}$” where $\mathcal{G}$ maybe was referring to the group and not the graph;
- “assignment of $g_i$ from Eq 2”, where it is not clear to me whether Eq 2 is the correct reference;
- “GWL that is only updates node colours” is not clear.

The paper lack a conclusion section. I would suggest moving it from the Appendix.

I believe this work is a good fit for the workshop bringing a theoretical contribution to this community.


**Submission Track:**

Extended Abstract (4 Page)

---

### Official Review · Reviewer_cdJR · 2022-10-13
**On the Expressive Power of Geometric Graph Neural Networks**

**Confidence:** 3
**Soundness:** 3
**Presentation:** 3
**Contribution:** 2
**Overall Rating:** 6

**Summary:**

The authors define a geometric Weiser-Lehman test for geometric graphs, aiming at extending the WL isomorphism test to geometric graphs up to relevant group action (permutation, rotation, translation). After defining the GWL test and Invariant GWL, the authors apply the test to inviariant and equivariant Graph Neural Networks to understand its equivariant and invariant properties.

**Questions:**

Could you extend the IGWL and GWL to other type of graphs (e.g. graphs with shapes on edges or nodes)?

**Limitations:**

The paper has a limited novelty in terms of methodology as it is briefly extending the WL to spatial graphs. However, the invariant and equivariant model are receiving a growing attention so a test could be useful for the community.

The authors should better address the limitation of the work as there is no conclusion or remarks or further development section in the paper. I would suggest to turn it into a proceeding as much of the content which is relevant for the reader is in the appendix.

**Recommended Decision:**

3: Accept

**Relevance:**

3: Solid fit

**Strengths And Weaknesses:**

The paper properly addresses the important issue of graph invariant and equivariant graph neural networks. The exposition should be improved, but overall address a valuable research question. I would change the name from geometric graph to spatial graphs (as the nodes have spatial attributes) or maybe geometric spatial graphs.


Premilinaries could be rewritten to make it easier to read. For example:
-	G-orbit space injective function could just be named as G invariant function as it makes the notation less heavy?
-	It is not clear to me the dimension of g(i) and consequently how the group action can act on it
-	Figure 1 middle panel, I think as you are showing the results for i-th node, at step 1 only the two neighbouring nodes should be colored and the other should be left gray.

Theorem 8 should be turned into a remark. As the authors state just after, it is a consequence of the fact that equivariant is a stronger property then invariant. It is worth sta

Even if you explain it in the appendix, I think a line should be added before talking about GNN in the paper. Which GNN are you talking about?

A plot about k-hop might help the reader understanding.

Titol 2.2 change what with which.


**Submission Track:**

Extended Abstract (4 Page)

---

### Official Review · Reviewer_jm6Q · 2022-10-15
**Simple yet significant analysis of expressivity in geometric graphs**

**Confidence:** 4
**Soundness:** 3
**Presentation:** 3
**Contribution:** 3
**Overall Rating:** 7

**Summary:**

Application of WL test to characterize the expressive power of GNNs for geometric graphs. The paper gives a detailed background and defines the Geometric WL test. It also gives an invariant GWL. Considering two cases of isomorphism in the graphs, the authors give propositions. The paper ends with a comparison of the expressive power of GWL and  IGWL.

**Questions:**

The work is theoretically sound. It would be interesting to look further at the $k$-WL for geometric graph neural networks.

**Limitations:**

1. Clarity in notations/symbols used would help the reader. Intuitive symbols would definitely help.
2. There are some small typographical errors; for example, geometric graph $\mathcal{G} = (\textbf{\textit{A}}, \textbf{\textit{S}}, \textbf{\textit{V}} \textbf{\textit{X}})$ where a comma is missing.

**Recommended Decision:**

3: Accept

**Relevance:**

3: Solid fit

**Strengths And Weaknesses:**

The work is a simple extension of WL to the geometric graphs. Seems significant in determining the expressive power of geometric graphs neural networks. Though it might not be previously attempted, it is an intuitive extension of WL. The submission lacks some clarity in some places regarding the terminology used.

**Submission Track:**

Extended Abstract (4 Page)

---

### Decision · Program_Chairs · 2022-10-21

Accept (Oral)